# Increased Oxygen Treatment in the Fermentation Process Improves the Taste and Liquor Color Qualities of Black Tea

**DOI:** 10.3390/foods14152736

**Published:** 2025-08-05

**Authors:** Xinfeng Jiang, Xin Lei, Chen Li, Lixian Wang, Xiaoling Wang, Heyuan Jiang

**Affiliations:** 1Jiangxi Economic Crops Research Institute, Jiangxi Provincial Key Laboratory of Plantation and High Valued Utilization of Specialty Fruit Tree and Tea, Nanchang 330006, China; jiangxinyue003@163.com (X.J.);; 2College of Biological and Environmental Engineering, Jingdezhen University, Jingdezhen 333000, China; 3Institute of Biological Resources, Jiangxi Academy of Sciences, Nanchang 333104, China; 4Institute of Urban Agriculture, Chinese Academy of Agricultural Sciences, Chengdu 610299, China

**Keywords:** black tea, oxygen treatment, fermentation, sensory quality, metabolomics

## Abstract

Black tea is widely consumed worldwide, and its characteristic taste and color result from fermentation, where polyphenols are enzymatically oxidized to generate major pigments, including theaflavins (TFs), thearubigins (TRs), and theabrownins (TBs). This study investigated the effects of increased oxygen treatment during fermentation on the flavor attributes and chemical properties of Congou black tea. Fresh tea leaves (variety “Fuyun 6”) were subjected to four oxygen treatments: 0 h (CK), 1 h (TY-1h), 2 h (TY-2h), and 3 h (TY-3h), with oxygen supplied at 8.0 L/min. Sensory evaluation revealed that oxygen-treated samples exhibited tighter and deeper-colored leaves, a redder liquor, fuller taste, and a sweeter fragrance compared with CK. Chromatic analysis showed significant increases in redness (a*) and luminance (L*), alongside reduced yellowness (b*), indicating enhanced liquor color. Chemical analyses demonstrated elevated levels of TFs, TRs, and TBs in oxygen treatments, with TRs showing the most pronounced increase. Non-targeted metabolomics identified 2318 non-volatile and 761 volatile metabolites, highlighting upregulated flavonoids, phenolic acids, and lipids, and downregulated catechins and tannins, which collectively contributed to improved taste and aroma. Optimal results were achieved with 2–3 h of oxygen treatment, balancing pigment formation and sensory quality. These findings can provide a scientific basis for optimizing oxygen conditions in black tea fermentation to improve product quality.

## 1. Introduction

Black tea is one of the most widely consumed beverages worldwide and is associated with antioxidant, antihypertensive, hypoglycemic, and cardioprotective properties [1]. Its manufacture involves withering, rolling, fermentation, and drying, among which fermentation is crucial because it determines the formation of major flavor components and accumulation of key aroma compounds [2]. During fermentation, polyphenols are enzymatically oxidized by polyphenol oxidase and peroxidase, producing characteristic pigments such as theaflavins (TFs), thearubigins (TRs), and theabrownins (TBs). These oxidation products largely determine the color, taste, aroma, and bioactive properties of black tea [3,4].

Fermentation is affected by multiple parameters, including temperature, oxygen availability, time, and humidity. Qu et al. reported that a temperature of 30–32 °C enhanced the sensory quality of black tea, whereas fermentation at 25–28 °C improved antioxidant activity, inhibited α-amylase and α-glucosidase, and reduced intestinal glucose absorption [5]. Chen et al. observed that high oxygen availability promoted the oxidation of catechins, flavonoid glycosides, and phenolic acids, while increasing glutamic acid accumulation, leading to reduced bitterness and astringency and enhanced umami taste [6]. Wang et al. found that a 3-h fermentation period improved simple catechin retention, converted bitter and astringent components, increased soluble sugar content, and promoted aroma compound formation in Yunnan Congou black tea [7]. Hua et al. suggested that fermenting at 25–30 °C for 60–90 min strongly influenced phenolic conversion and reaction pathways. Longer fermentation times reduced liquor brightness, whereas moderate low temperatures preserved polyphenol oxidase activity and improved liquor color [8]. Humidity, expressed as relative humidity in the fermentation environment, plays a key role, as water acts both as a solvent and a reactant in fermentation reactions. Variations in humidity alter leaf moisture content by affecting water evaporation, thereby influencing chemical transformations and the quantity of fermentation products [9]. Few studies have systematically investigated humidity regulation, and fermentation humidity in commercial tea production is often adjusted empirically without a strong scientific basis [10].

Black tea is broadly classified as Congou or broken black tea based on leaf shape, which is determined by differences in rolling methods. Congou black tea is rolled into strips using traditional rolling machines, whereas broken black tea is rolled and cut into granules. The rolling/cutting process in broken black tea causes extensive cell rupture, increases the exposed surface area, and accelerates chemical reactions such as polyphenol oxidation and glycoside hydrolysis, allowing fermentation to be completed within 1–2 h [11]. In contrast, Congou black tea ferments for 3–6 h due to slower oxidation [12]. Because of these differences, oxygen-enriched fermentation is expected to have a greater impact on Congou black tea, although detailed evidence remains limited. This study aimed to evaluate the effect of increased oxygen exposure during fermentation on Congou black tea. Different oxygenation times were tested, and their influence on taste and liquor color was analyzed to establish a scientific basis for improving Congou black tea quality.

## 2. Materials and Methods

### 2.1. Tea Sample Processing

Fresh clonal shoots of the “Fuyun 6” variety, each comprising one bud with two leaves, were gathered in early June 2023 from the Ecological Tourist Tea Plantation Base of the Jiangxi Cash Crops Research Institute in Nanchang, China. The collected material was homogenized and evenly spread in a 6CWD-200 withering trough (Zhejiang SunYong Machinery Co., Ltd., Quzhou, China). Withering commenced with 1 h of cold air treatment, followed by warm air exposure at 32 °C until the moisture level of the leaves declined to roughly 58%. After rehydration, the leaves underwent mechanical rolling for 2 h using a 6CR-45 roller (Zhejiang SunYong Machinery Co., Ltd.), applying a pressure cycle of light–heavy–light. Rolled leaves were subsequently loosened and mixed thoroughly. Fermentation was carried out in a 6CHFJ-5B artificial climate chamber (Zhejiang Hong Wu Huan Tea Equipment Co., Ltd., Quzhou, China) at 32 °C under four oxygenation durations: 0 h (CK), 1 h (TY-1h), 2 h (TY-2h), and 3 h (TY-3h). Each treatment included three replicates, with 1 kg of leaf material per replicate. The entire fermentation process lasted 210 min, with constant airflow maintained through the chamber’s built-in ventilation system. After fermentation, the tea leaves were dried using a 6CTH-8.0 dryer machine (Zhejiang Zhu Feng Machinery Co., Ltd., Quzhou, China) at an initial drying time of 10 min at 110 °C, a redrying time of 30 min at 90 °C, and a final drying at 80 °C for 60 min, with sufficient cooling to room temperature and rewetting between each drying step (Figure 1). The dried tea was stored at 20 °C in a freezer until further analysis.

### 2.2. Oxygenation Treatment Method in Congou Black Tea Fermentation Process

A pre-designed oxygen-supplied fermentation device was designed as a hermetically sealed system. Industrial-grade oxygen cylinders were employed as the oxygen source, with oxygen delivered from the bottom of the device, with an oxygen supply rate of 8.0 L/min. Rolled tea leaves were placed into the oxygen-supplied fermentation device, with the tea pile covered by a damp cloth. The entire oxygenated fermentation device was installed in a traditional fermentation chamber and maintained under the same ambient temperature and relative humidity conditions, the same as conventional fermentation processes.

### 2.3. Chemical Reagents

Analytical-grade methanol, acetonitrile, glacial acetic acid, and formic acid were supplied by Thermo Fisher Scientific (Waltham, MA, USA). Reference standards for catechins and theaflavins were sourced from Shanghai Yuanye Biotechnology Co., Ltd. (Shanghai, China). Ultrapure water (resistivity 18.2 MΩ·cm) was prepared using a Milli-Q purification system (Millipore, Billerica, MA, USA). Additional reagents and solvents were purchased from China National Pharmaceutical Group Corporation (Beijing, China) and met analytical-grade specifications.

### 2.4. Sensory Evaluation

Sensory evaluation was performed in accordance with GB/T 23376–2018 and GB/T 14487–2017, the Chinese national standards for tea sensory methodology and vocabulary. Congou black tea samples were homogenized and randomized before assessment. Coded dry-leaf samples were evaluated for visual characteristics. For each evaluation, 3 g of tea was brewed with 150 mL of boiling water in standard cupping vessels, covered, and steeped for 5 min. The infusion was transferred to white porcelain bowls, and assessments of aroma and taste were conducted at 30 s intervals. A panel of five trained tasters independently scored each sample using a 100-point scale: 25% for dry-leaf appearance, 10% for liquor color, 25% for aroma, 30% for taste, and 10% for infused-leaf quality [13]. To ensure consistency, three certified assessors verified the sensory attributes. Afterwards, the appearance, aroma, liquor color, taste, and brewed leaves were evaluated by three experienced assessors authenticated by professional organizations.

### 2.5. Chemical Composition Determination

Water-extractable content was quantified following GB/T 8305–2013. Exactly 2.0 g of finely ground tea was allowed to infuse in 500 mL of boiling water for 45 min, with agitation every 10 min. The extract was obtained after filtration, and the solid residue was oven-dried at 120 °C for 2 h. The content of water-soluble compounds was calculated as the percentage of mass lost during extraction:$$x=(1-\frac{m_{1}}{M_{0}\times m})\times 100\%$$
where m_1_ represents the mass of tea residue (g); M_0_ represents the mass of the specimen (g); and m is the percentage of dry materials.

Total polyphenol content was measured according to “Determination of total polyphenols and catechins content in tea” (GB/T 8313–2018). Tea powder (0.2 g) was extracted twice with 5 mL of 70% methanol (*v*/*v*) at 70 °C for 10 min per extraction. Each extract was centrifuged at 3500 rpm for 10 min (5810R, Eppendorf, Hamburg, Germany), and the resulting supernatants were combined and diluted 100-fold with distilled water. A 1 mL aliquot of this solution was reacted with 5 mL of 10% Folin–Ciocalteu reagent (*v*/*v*). After 8 min, 4 mL of 7.5% sodium carbonate (*w*/*v*) was added. The mixture was incubated for 60 min, and absorbance was read at 765 nm using a Shimadzu UV-2550 spectrophotometer (Kyoto, Japan) with a 10 mm cuvette. Gallic acid served as the calibration standard, and distilled water was used as a blank. Free amino acid content was determined via ninhydrin colorimetry based on GB/T 8314–2013. A 3.0 g sample was extracted in 450 mL of boiling water (100 °C) for 45 min, with agitation every 10 min. The filtrate was collected, and the residue was rinsed with additional hot water. The combined extracts were cooled and brought to 500 mL with distilled water. For analysis, 1 mL of the extract was mixed with 0.5 mL of phosphate buffer (pH 8.0) and 0.5 mL of 2% ninhydrin (*w*/*v*). The solution was heated in a boiling water bath for 15 min, then diluted to 25 mL with distilled water and allowed to stand for 10 min. Absorbance was recorded at 570 nm in a 10 mm cuvette. Theanine was used to generate the standard curve, with distilled water as the blank.

### 2.6. Determination of Caffeine, TF, TR, and TB Concentrations

Total flavonoid concentration was assessed using a colorimetric approach based on complex formation with aluminum chloride. For pigment compound separation, TFs, TRs, and TBs were successively extracted using ethyl acetate, *n*-butanol, and ethanol, respectively. Quantification followed established analytical protocols [13]. Tea infusions prepared as per sensory testing procedures were used to evaluate liquor coloration, which was measured with a CS-820N spectrophotometer (Hangzhou Caipu Technology Co., Ltd., Hangzhou, China).

Theaflavins, catechins, and caffeine were identified and quantified by HPLC (1260 Infinity, Agilent Technologies Co., Ltd., Santa Clara, CA, USA). TF analysis was conducted under the GB/T 31740.3-2015 guidelines using an Agilent TC-C18 column (250 mm × 4.6 mm, 5 μm). The chromatographic parameters included: flow rate, 0.7 mL/min; column temperature, 35 °C; injection volume, 5 μL; detection wavelength, 278 nm. Two mobile phases were used as follows: mobile phase A composed of 90 mL acetonitrile with 20 mL glacial acetic acid and 2 mL EDTA-2Na (10 mg/mL), with ultrapure water fixed to 1000 mL; mobile phase B composed of 800 mL acetonitrile with 20 mL glacial acetic acid and 2 mL EDTA-2Na (10 mg/mL). Catechins and caffeine were measured using the same column setup, in accordance with GB/T 8313-2018. For this analysis, the mobile phases were as follows: mobile phase A constituted by ultrapure water (containing 0.1% formic acid); mobile phase B composed of methanol (containing 0.1% formic acid). The gradient profile for separation of theaflavins, catechins, and caffeine was 0–10 min, 100% A; 10–25 min, 68% A and 32% B; 25–35 min, 100% A.

### 2.7. Chromatic Aberration Analysis of Tea Liquors

Tea liquor was obtained using the same protocol as the sensory assessment. Color attributes were measured using a CR-400 colorimeter (Konica Minolta, Tokyo, Japan) according to the CIELAB color space. For each sample, L* (lightness), a* (red-green), and b* (yellow-blue) values were recorded across 20 data points. All measurements were performed in triplicate.

### 2.8. Analysis of Non-Volatile Metabolites in Black Teas Using UPLCMS/MS

#### 2.8.1. Extraction of Non-Volatiles

For each sample, 100 mg of tea powder was extracted in 1.2 mL of 70% methanol. Vortexing was performed for 30 s at 30 min intervals over six cycles. The mixture was then kept at 4 °C overnight. After incubation, samples were centrifuged at 16,743 *g* for 10 min. The resulting supernatants were passed through 0.22 μm membrane filters (SCAA-104, ANPEL, Shanghai, China) to remove particulates. All extracts were analyzed in triplicate using ultra-performance liquid chromatography coupled with electrospray ionization tandem mass spectrometry (UPLC–ESI–MS/MS) for non-volatile metabolite profiling.

#### 2.8.2. UPLC Conditions and ESI-MS/MS

Metabolomic profiling was performed by MetWare (Wuhan, China) [14] using a Shimadzu Nexera X2 UPLC system (Kyoto, Japan) connected to an AB Sciex 4500 QTRAP mass spectrometer (Thermo Fisher Scientific, Waltham, MA, USA). Chromatographic separation was conducted on an Agilent SB-C18 column (2.1 × 100 mm, 1.8 μm). The mobile phase consisted of solvent A (water containing 0.1% formic acid) and solvent B (acetonitrile containing 0.1% formic acid). The gradient elution program was as follows: initial conditions of 95% A and 5% B were gradually shifted to 5% A and 95% B over 9 min, maintained at this composition for 1 min, returned to 95% A and 5% B within 1.1 min, and held for 2.9 min. The flow rate was 0.35 mL/min, the column temperature was maintained at 40 °C, and the injection volume was 4 μL. Eluted compounds were ionized using an electrospray ionization (ESI) source in both positive and negative modes. The mass spectrometer operated in both triple quadrupole (QQQ) and linear ion trap (LIT) scan modes, with data acquisition managed using Analyst 1.6.3 software (AB Sciex). ESI source settings were as follows: source temperature, 550 °C; ion spray voltage, +5500 V in positive mode and −4500 V in negative mode; gas I, 50 psi; gas II, 60 psi; curtain gas, 25 psi. Collision-activated dissociation was applied at high energy. Instrument calibration was performed using 10 and 100 μmol/L polypropylene glycol standards for QQQ and LIT modes, respectively. Metabolite quantification employed multiple reaction monitoring (MRM) with nitrogen as the collision gas at medium intensity. Declustering potential (DP) and collision energy (CE) were optimized for each MRM transition based on metabolite-specific retention times.

### 2.9. Analysis of Volatile Metabolites in Black Teas Using GC-MS

#### 2.9.1. Extraction of Volatiles

Volatile extraction followed a modified version of a previously reported method [15]. A 600 mg portion of tea sample was weighed and combined with 500 mg of NaCl in a 20 mL headspace vial. Ten milliliters of boiling water was added to infuse the tea. A polydimethylsiloxane (PDMS) Twister (Gerstel, Germany; 10 mm length, 1.0 mm thickness, 24 μL capacity) was then immersed in the infusion to adsorb volatiles. Extraction was conducted at 80 °C with agitation at 1200 rpm for 30 min using a multi-position magnetic stirrer (SP200-2T, Miu Instruments Co., Ltd., Hangzhou, China). Each sample was prepared in triplicate.

#### 2.9.2. Thermal Desorption

Volatile-laden stir bars were subjected to thermal desorption using a Gerstel TDU system. The desorption chamber was held at 30 °C for 1 min, then rapidly heated to 240 °C at 100 °C/min and maintained at that temperature for 5 min. The cooled injection system (CIS-4 PTV, Gerstel) was pre-chilled to −100 °C using 99.99% pure liquid nitrogen and held for 1 min before being ramped to 280 °C at 12 °C/s and maintained for 3 min to ensure complete desorption.

#### 2.9.3. GC–MS Analysis

Volatile compounds were separated and identified using an Agilent 7890B gas chromatograph equipped with an Agilent 5977B mass selective detector (Agilent Technologies, Santa Clara, CA, USA). Separation was performed on an HP-5MS column (30 m × 0.25 mm, 0.25 μm film thickness) with helium (>99.99%) as the carrier gas at a constant flow rate of 1.6 mL/min. The oven temperature program was as follows: initial temperature of 50 °C held for 2 min, ramped to 170 °C at 4 °C/min and held for 5 min, then increased to 265 °C at 10 °C/min and held for a final 5 min. The mass spectrometer operated in electron impact (EI) mode at 70 eV. The ion source and transfer line were maintained at 220 °C and 280 °C, respectively. Spectra were acquired across a scan range of 30–600 amu.

#### 2.9.4. Qualitative and Quantitative Analysis of Volatiles

Compound identification was performed using the NIST 2014 mass spectral library. Retention indices (RIs) determined experimentally were compared with theoretical RIs of n-alkane standards (C8–C40). Compounds were considered reliably identified when RI deviations were within ±20. Quantification was based on relative peak area obtained from GC–MS total ion chromatograms.

### 2.10. Statistical Analysis

Quantitative assays for water extract, total polyphenols, free amino acids, caffeine, and catechins were performed in triplicate. Results were reported as mean ± standard deviation (SD, n = 3). Treatment effects were assessed using one-way analysis of variance (ANOVA), followed by Fisher’s least significant difference (LSD) test to identify statistically significant differences at *p* < 0.05. Statistical computations were carried out in SPSS Statistics 26.0. Graphs were prepared using Origin 2021 and Adobe Illustrator 2021.

Metabolomic data from UPLC–MS/MS and GC–MS were processed using the MetWare Cloud platform (https://cloud.metware.cn/, 10 July 2023). Analyses included principal component analysis (PCA), orthogonal partial least squares discriminant analysis (OPLS-DA), K-means clustering, hierarchical clustering, and temporal trend profiling of volatile metabolites associated with aroma.

## 3. Results and Discussion

### 3.1. Effects of Different Treatments on Sensory Quality of Black Tea

The sensory evaluation results for the four black tea samples are presented in Figure 2 and Appendix A. Photographs of the tea appearance, liquor, and brewed leaves are also shown in Figure 2 to illustrate visual differences among treatments.

Relative to the CK, TY-1h, TY-2h, and TY-3h exhibited a tighter and darker appearance, redder liquor, thicker taste, sweeter aroma, and redder infused leaves. Taste scores for TY-1h, TY-2h, and TY-3h were 87.0, while liquor color scores for the three treatments reached 91.0 (Figure 2). These findings indicate that increased oxygen exposure during fermentation (TY-1h, TY-2h, and TY-3h) enhanced both taste and liquor color quality in black tea.

### 3.2. Effects of Different Treatments on the Chromatic Aberration of Tea Liquor

As illustrated in Figure 3 and Appendix A, the a* value (indicating redness) and L* value (indicating brightness) increased significantly from CK to TY-3h, whereas the b* value (representing yellowness) gradually decreased. Compared with CK, the values of a* TY-1h, TY-2h, TY-3h were further increased (20.88, 28.90, and 36.70%, respectively) and the value of L* was further increased (1.27, 2.18, and 2.50%, respectively) (Figure 3). Some studies have shown that a* value, b* value, and the total sensory score of black tea were significantly or very significantly positively correlated, indicating that the higher the degree of red and yellow color of black tea broth, the higher the sensory score, while the brightness of black tea broth L* was significantly negatively correlated with the total sensory score, that is, within a certain range of the black tea broth the L* value is small, the red TBs and the yellow TFs of the broth are rich, and the sensory score of the tea broth is high [16]. These findings further confirmed that increased oxygen exposure during fermentation improved the liquor color quality of black tea.

### 3.3. Effects of Different Treatments on the General Biochemistry Components

Tea taste and liquor color are strongly associated with non-volatile constituents [17]. Therefore, major biochemical components were analyzed. Water extracts, which represent the total soluble components in hot water and are positively associated with taste thickness, were measured. The water extract content was 46.07% in CK, 46.42% in TY-1h, 44.16% in TY-2h, and 43.90% in TY-3h. Significant differences (*p* < 0.05) were observed between CK and TY-1h compared with TY-2h and TY-3h (Figure 4). Tea polyphenol content was 12.52% in CK, 11.66% in TY-1h, 6.81% in TY-2h, and 6.72% in TY-3h, with significant differences (*p* < 0.05) between CK and TY-1h versus TY-2h and TY-3h (Figure 4). Total amino acid levels were 3.03% in CK, 2.90% in TY-1h, 3.01% in TY-2h, and 2.97% in TY-3h. TY-2h exhibited significantly lower amino acid content than CK, indicating a negative correlation between total amino acid concentration and extended oxygen treatment time [18]. However, compared with TY-1h, amino acid levels in TY-2h and TY-3h increased significantly, returning to levels comparable to CK (Figure 4). TFs, TRs, and TBs are enzymatic oxidation products of polyphenols [19]. Their concentrations increased progressively from CK to TY-3h, with TRs showing the most pronounced increase, followed by TFs and TBs (Figure 3). This suggests that prolonged oxygen treatment strongly promotes TR formation, with comparatively lower effects on TFs and TBs.

The fermentation mechanism of black tea is mainly an enzyme-driven catechin oxidation polymerization reaction, and the oxidation rate of catechins reflects the formation of black tea quality [20]. It was shown that the oxidation of catechin components in the fermentation process was a primary reaction, the correlation coefficients of the oxidation kinetic curves were all greater than 0.95, and the catechin oxidation rate constants (k) were in the order of EGC > EGCG > ECG > EC. The oxidation power of the catechins in the natural fermentation process was lost by 89.73% after 6 h, and there was a lack of fermentation power, which showed that 6~8 h was the preferred time for the natural fermentation process of artisanal black tea. This indicates that 6~8 h is the ideal time for the natural fermentation of black tea. Therefore, 6~8 h is the ideal time for the natural fermentation process of black tea. In this experiment, aerobic fermentation with controlled temperature and humidity was used, so 3~4 h was the ideal time for the fermentation process of black tea. The oxidation kinetics of the catechin components varied, the oxidation rate of simple catechin components (EGC, EC) showed an undulating pattern, while the oxidation rate of ester-type catechin components (EGCG, ECG) basically maintained a gradual decrease trend, which was mainly attributed to the different chemical properties of different components of the catechins and the different chemical reactions they were involved in [21]. These results indicate that the extended TY-3h treatment had the strongest impact on the formation of catechins, particularly EGC, EGCG, ECG, and EC.

### 3.4. Effect of Increased Oxygen Treatment on Non-Targeted Metabolomics Analysis Based on UPLC-MS

#### 3.4.1. Composition and Variation Trend of Non-Volatile Metabolites Across Four Different Treatments

To assess the impact of increased oxygen exposure on black tea composition, non-targeted metabolomic profiling was conducted to compare non-volatile and volatile constituents across the four treatments. Using UPLC–MS/MS combined with an in-house library, metabolite identification was based on authentic standards, MS^2^ fragmentation patterns, curated databases, and the prior literature [22,23]. A total of 2318 metabolites spanning 12 major classes were identified. These included 549 flavonoids, 419 phenolic acids, 189 amino acids and derivatives, 187 lipids, 175 alkaloids, 119 terpenoids, 117 organic acids, and 110 lignans and coumarins. Additionally, 80 nucleotide-related compounds, 78 tannins, 15 quinones, and 280 metabolites classified as “others” were detected (Appendix A).

PCA was performed to evaluate metabolite differences among black teas subjected to varying fermentation durations. The first two principal components (PC1 and PC2) accounted for 44.84% of the total variance (PC1: 32.87%, PC2: 10.97%). In the PCA score plot, TY-treated tea samples clustered separately, indicating marked changes in metabolite profiles with extended oxygen treatment (Figure 5A). Meanwhile, the samples of the four treatments were arranged in sequence in the PC1 direction, indicating that the differences among the sample groups were mainly affected by the oxygen supply time.

The results of OPLS-DA analysis indicated that the first two components explained 32.6 and 10.2% of the variances, respectively, and there was obvious separation among the four tea samples, which was consistent with the results of PCA (Figure 5B). After 200 permutations, we obtained the vector values of R2Y = (0.0, 1) and Q2 = (0.0, 0.941). These results indicate that the model is not overfitted and shows a high level of reliability (Figure 5C).

#### 3.4.2. Characterization of Differential Non-Volatile Metabolites of Four Different Treatments

To gain a deeper insight into the different treatments in non-volatile metabolic compounds with oxygen supply time, we compared the tea samples with different oxygen supply time with CK. The VIP values were calculated by establishing between-group comparisons of OPLS-DA models (the model scores and diagnostic results can be found in Appendix A), and screening was conducted in combination with the significance test (*p* value). The screening criteria were VIP ≥ 1 and *p* < 0.05.

After screening, 366, 528, and 695 differential metabolites were identified in the TY-1h_vs._CK, TY-2h_vs._CK, and TY-3h_vs._CK comparisons, respectively. Interestingly, the types of differential metabolites increased with the increase in oxygen supply time. The difference in metabolites was most significant in TY-3h, indicating that, among the four treatments, the improvement effect of TY-3h on quality was the most different from that of CK (Figure 5D). Venn diagram analysis of the three comparison groups further confirmed this observation result, with unique metabolites across comparisons following the order: TY-1h vs. CK (61), TY-2h vs. CK (67), and TY-3h vs. CK (233). Furthermore, there were 223 differential metabolites identified in four groups.

These metabolites can reflect the impact of oxygen fermentation treatment on the metabolic profile of black tea. As shown in Figure 5D, flavonoids account for the largest proportion among differential metabolites across the four tea samples, accounting for 29.73% of the total. Phenolic acids represented the second major component (18.92%), followed by lipids (12.61%) and terpenoids (8.11%), with all remaining categories each contributing less than 5%.

The contents of 223 common differential metabolites in different treatments were subjected to HCA clustering and visualization, with visualization results presented in Figure 5F. The column-wise clustering pattern revealed that replicate samples from identical treatments consistently grouped together, demonstrating robust reproducibility of the metabolomic profiling. Interestingly, the changing trends of the contents of these substances can roughly be divided into two distinct clusters: upward or downward accumulation patterns in the TY group.

#### 3.4.3. Key Differential Non-Volatile Metabolites of Four Different Treatments

Using a fold change threshold of ≥5 or ≤0.2, 222 non-volatile metabolites showing significant differences were identified. These metabolites play key roles in determining tea quality and were mainly classified as flavonoids, phenolic acids, and terpenoid derivatives (Figure 6). Among phenolic acid metabolites, methyl hydroxycinnamate, p-coumaric acid methyl ester, and methyl caffeate exhibited notably higher levels across all treatments. In the category of nucleotides and derivatives, 9-(arabinosyl)hypoxanthine, cyclic 3′,5′-adenylic acid, and guanosine 3′,5′-cyclic monophosphate showed marked variations among the different oxygen treatments.

The total of non-volatile differential metabolites was analyzed using K-means clustering, with K = 2 to identify the metabolites accumulated in the upregulation and downregulation of the TY groups, and two subclusters were obtained. Subcluster 1 contained 99 metabolites and was upregulated and accumulated in the TY treatment, while subcluster 2 contained 123 metabolites and was downregulated and accumulated in the TY treatment (Figure 6A,B).

The upregulated metabolites in this study primarily encompass flavonoids, lipids, phenolic acids, terpenoids, amino acids and their derivatives, and nucleotides and related compounds. Flavonoid compounds are of paramount importance not only to the taste quality but also to the chromatic characteristics of tea leaves. The current investigation identified 13 upregulated flavonoid compounds in increased oxygen treatment, comprising chalcones (4), flavanones (1), flavones (5), and flavonols (3). Flavonols predominantly exist in glycosylated forms within tea leaves, including notable compounds such as quercetin-4′-O-glucoside(spiraeoside) *, kaempferol-3-O-(2″-p-coumaroyl)glucoside, and quercetin-3-O-(2″-O-caffeoyl)glucoside-(1→2)-(6″-malonyl)glucoside. These compounds impart a delicate slightly astringent flavor profile while potentiating the inherent bitterness of caffeine in tea infusions. The TY treatment also induced significant accumulation of five flavones, including 6-hydroxyluteolin 5-glucoside *, apigenin (4′,5,7-trihydroxyflavone), chrysoeriol-5-O-glucoside, chrysoeriol-6-C-glucoside-7-O-glucoside, and apigenin-4′-O-(2″,6″-di-O-p-coumaroyl)glucoside. The majority of these phytochemicals exhibit glycosylation patterns, a characteristic structural feature that enhances their solubility and bioavailability in aqueous infusion systems.

The upregulated lipid profile primarily consists of various free fatty acids alongside lysophosphatidylcholines (LPCs) and lysophosphatidylethanolamines (LPEs), compounds recognized for their roles in membrane remodeling and flavor precursor formation.

Phenolic acids, crucial enhancers of umami taste and architects of tea’s characteristic flavor profile [24], demonstrated dynamic accumulation patterns. Notably, 4,6-(S)-hexahydroxydiphenoyl-D-glucose and chicoric acid peaked in TY-2h samples, while other phenolic acids—including 3-methylsalicylic acid, 3-hydroxycinnamic acid *, brevifolin carboxylic acid, octyl gallate, phenethyl caffeate, 1-O-rhamnose-3-O-caffeoyl quinic acid, and 1,2,6-tri-O-galloyl-β-D-glucose *—exhibited progressive accumulation proportional to aerobic fermentation duration.

Terpenoid metabolism featured upregulated diterpenoids and sesquiterpenoids, secondary metabolites known to contribute to tea’s aromatic complexity. Amino acid derivatives manifested predominantly as dipeptides—glycylphenylalanine *, Phe-Ser, Phe-Thr, Glu-Phe, and γ-glutamylphenylalanine—likely hydrolysis products of proteolytic activity during fermentation. Concurrently, the sustained upregulation of nucleotide derivatives, including 9-(arabinosyl)hypoxanthine, cyclic 3′,5′-adenylic acid, guanosine 3′,5′-cyclic monophosphate, guanosine, hypoxanthine, and guanine, suggests potential associations with cellular structural disruption and macromolecular nucleic acid hydrolysis. These metabolic shifts collectively reflect the biochemical orchestration underlying tea fermentation’s transformative effects [20].

The downregulated metabolites predominantly feature flavonoids, phenolic acids, tannins, and terpenoids. The diminished flavonoid profile encompasses various flavanols, anthocyanins, flavones, flavanones, flavonols, and dihydroflavonols. Notably, multiple catechin derivatives exhibited significant downregulation, including catechin, gallocatechin, epicatechin gallate, gallocatechin gallate, epigallocatechin 3-O-(3-O-methyl)gallate, epigallocatechin 3-O-cinnamate, epigallocatechin 3,5-digallate, 3′-O-methyl-6-hydroxygallocatechin 3-O-(N-ethylglutamine ester) 3′-gallate, 4′-O-methyl-6-hydroxygallocatechin 3-O-(N-ethylglutamine ester) 3′-gallate, catechin-catechin-catechin, and gallocatechin-(4α→8)-catechin-(4α→8)-catechin. Tannin reduction manifested through decreased proanthocyanidins such as procyanidin B2, procyanidin B4, procyanidin C1, and procyanidin C2, concomitant with a marked reduction in sucrose content.

These metabolic changes indicate extensive polymerization of simple flavonoids during aerobic fermentation, generating darker-hued organoleptically complex polymeric structures. This biochemical transformation critically contributes to the characteristic “red liquor and red leaves” quality parameters of black tea. The coordinated downregulation of phenolic acids and tannins further facilitates flavor refinement, reducing astringency while enhancing the mellow full-bodied taste profile through selective molecular reorganization [25].

### 3.5. Effect of Increased Oxygen Treatment on Non-Targeted Metabolomics Analysis Based on GC-MS

#### 3.5.1. Composition and Variation Trend of Volatile Metabolites of Four Different Treatments

Our investigation into oxygen-permeable fermentation’s impact on black tea flavor profiles employed untargeted metabolomics to characterize volatile organic compounds (VOCs). The analysis identified 761 volatile metabolites spanning 16 categories, including 158 terpenoids, 130 esters, 113 heterocyclic compounds, 66 alcohols, 49 aldehydes, 45 aromatic hydrocarbons, and 7 aliphatic hydrocarbons, with remaining categories each constituting <5% of total detected VOCs (Appendix A).

PCA revealed two principal components accounting for 73.75% of total variance (PC1: 56.83%, PC2: 16.92%). Mirroring non-volatile metabolome findings, PCA score plots demonstrated distinct intra-group clustering of TY tea samples, with four experimental treatments forming sequential distribution along the PC1 axis (Figure 7A). OPLS-DA showed components explaining 55.8% and 15.7% variance, respectively. Permutation testing (200 iterations) validated model robustness with vector values R^2^Y = (0.0, 0.989) and Q^2^ = (0.0, 0.977) (Figure 7B,C), indicating significant metabolic differentiation among experimental groups.

#### 3.5.2. Key Differential Volatile Metabolites of Four Different Treatments

Following rigorous screening by variable importance projection (VIP > 1) and statistical significance threshold (*p* < 0.05), comparative analyses revealed 168, 356, and 496 differential volatile metabolites in TY-1h_vs._CK, TY-2h_vs._CK, and TY-3h_vs._CK, respectively (Figure 7D), exhibiting a progressive increase corresponding to extended aeration duration. Unique metabolic signatures were identified across comparisons, comprising 7, 26, and 165 treatment-specific volatiles in respective groups, demonstrating substantial aromatic modification induced by TY-3h intervention.

Flavor profiling of differential volatile metabolites across comparative groups was systematically executed, prioritizing the top 10 organoleptic categories exhibiting maximal annotated compound diversity. As illustrated in Figure 8A, the TY-1h_vs._CK comparison revealed predominant organoleptic signatures of greenness, fruitiness, and apple-like notes, complemented by subsidiary sweet, banana, vinous, and fresh undertones. The TY-2h_vs._CK group demonstrated substantial expansion in both qualitative and quantitative dimensions, with reinforced green/sweet duality, intensified apple nuances, and diversified fruity complexity. Temporal progression to TY-3h_vs._CK manifested sensory recalibration toward sophisticated aroma spectra dominated by fruity/sweet convergence, floral prominence, and pear/apple synesthesia. This oxygenation-dependent phytochemical evolution illustrates a paradigm shift in black tea’s aromatic architecture—transitioning from simple vegetative notes to multifaceted bouquet development characterized by escalating fruity esters, sacchariferous derivatives, and floral terpenoid synergies during prolonged fermentation [26].

Venn analysis further delineated 164 conserved differential metabolites across all three treatments. Detailed classification of these shared metabolites (Figure 7E) revealed predominant chemical families: esters (20.25%), heterocyclic compounds (16.56%), terpenoids (16.56%), alcohols (9.82%), hydrocarbons (9.2%), ketones (7.98%), and aromatics (5.52%), with remaining categories collectively representing 13.21% of the total.

Hierarchical clustering analysis of pivotal volatile constituents revealed predominant downregulation patterns in alcohol, ester, heterocyclic compound, hydrocarbon, and terpenoid accumulation with prolonged aeration (Figure 7F). Notably, select aldehyde, ester, ketone, and terpenoid derivatives exhibited upregulated deposition profiles, potentially serving as critical contributors to the final black tea’s signature aromatic bouquet through their distinctive accumulation dynamics [26].

K-means clustering delineated 49 upregulated and 114 downregulated volatile metabolites (Figure 9A,B). Among the enriched constituents, we identified notable aromatic compounds including butanoic acid, 1-ethenylhexyl ester; 2H-pyran-3-ol, 6-ethenyltetrahydro-2,2,6-trimethyl-; ethanone, 1-(3,5-dimethylpyrazinyl)-; 2-((3,3-dimethyloxiran-2-yl)methyl)-3-methylfuran; and bicyclo[2.2.1]heptan-2-ol, 1,7,7-trimethyl-, (1S-endo)-, which demonstrated progressive accumulation patterns positively correlated with oxygenation duration. These pharmacologically active volatiles collectively impart signature organoleptic properties to black tea, particularly enhancing floral nuances, fruity undertones, and roasted aromatic complexity through their synergistic biosynthetic regulation (Figure 9C).

Progressive oxygenation duration elicited marked depletion of herbaceous volatiles, including 3-hexen-1-ol (Z)-, 3-hexen-1-ol (E)-, butanoic acid 3-hexenyl ester (Z)-, 3-hexen-1-ol acetate (E)-, and pantolactone. Concomitantly, citrus-associated aroma constituents α-terpinyl acetate and geranyl acetate demonstrated diminishing accumulation profiles. Intriguingly, signature black tea volatiles methyl salicylate and hexyl acetate showed significant depletion during aeration, presumably attributable to sensory quality recalibration favoring intensified saccharine and floral dominance in oxygen-fermented tea leaves (Figure 9C).

## 4. Conclusions

Comparative analysis revealed oxygenation treatment significantly accelerated enzymatic browning kinetics and chromatic transition toward characteristic russet pigmentation in Congou black tea fermentation, concurrently enhancing signature aroma development while abbreviating processing duration.

During initial phases, aerated fermentation achieves accelerated biosynthesis and supra-optimal accumulation of TFs and thearubigins TRs versus conventional anaerobic protocols. Paradoxically, sustained oxygenation beyond critical temporal thresholds induces accelerated TF/TR degradation kinetics exceeding natural metabolic clearance rates, compounded by disproportionate TB formation through oxidative polymerization. This dual-phase oxygenation effect culminates in detrimental effects on final product quality, manifesting as imbalanced astringency modulation and compromised liquor brightness.

Quantitative analysis of theaflavins, catechins, caffeine, and amino acids in tea samples with different fermentation degrees (I, II, III) reveal that during fermentation, total theaflavin content slightly increases while catechin levels decreases drastically, though the rate of decrease gradually slows. This phenomenon may result from the strong correlation between catechin reduction and theaflavin formation [27].

During black tea fermentation, catechins undergo enzymatic oxidation with polyphenol oxidase to yield substantial TFs. As fermentation progresses, the diminished catalytic capacity of polyphenol oxidase coupled with diminishing phenolic substrates progressively attenuates the rate of catechin depletion.

The overall reduction in free amino acids likely stems from their transformation during fermentation through three primary pathways. First, enzymatic decarboxylation and deamination convert amino acids into aromatic compounds like aldehydes, alcohols, and acids. Second, they participate in condensation reactions with polyphenols and carbohydrates to form quinones, aldehydes, acids, and alcohols. Third, amino acids interact synergistically with theaflavins and thearubigins, yielding dark-colored highly polymerized compounds [28,29].

Throughout the fermentation spectrum, distinct biochemical trajectories define tea quality. Samples subjected to light fermentation (I) retain elevated catechin levels, imparting an astringent and herbaceous infusion. Moderately fermented batches (II) exhibit diminished catechins alongside amplified theaflavin synthesis, culminating in a robust and refreshing character. Conversely, heavily fermented variants (III) demonstrate minimal catechin preservation with diminished infusion concentration yet paradoxically achieve a mellow and harmonious flavor profile through advanced oxidative polymerization.

Our comprehensive metabolomic profiling identified 761 volatile compounds spanning 16 distinct classes, revealing dynamic aromatic evolution under progressive oxygenation. Prolonged oxygenation duration correlated positively with both quantitative expansion of differential volatile metabolites and qualitative diversification of fragrant constituents, orchestrating sensorial metamorphosis toward fruity/sweet/floral dominance. Notably, oxygenation-driven biosynthesis manifested in gradual elevation of signature aromatic compounds including butanoic acid, 1-ethenylhexyl ester; 2H-pyran-3-ol, 6-ethenyltetrahydro-2,2,6-trimethyl-; ethanone, 1-(3,5-dimethylpyrazinyl)-; 2-((3,3-dimethyloxiran-2-yl)methyl)-3-methylfuran; and bicyclo[2.2.1]heptan-2-ol, 1,7,7-trimethyl-, (1S-endo)-. These oxygen-responsive phytochemicals demonstrated incremental elevation profiles, collectively contributing to the organoleptic evolution toward sophisticated fruity/floral bouquets characteristic of premium black teas.

## Figures and Tables

**Figure 1 foods-14-02736-f001:**
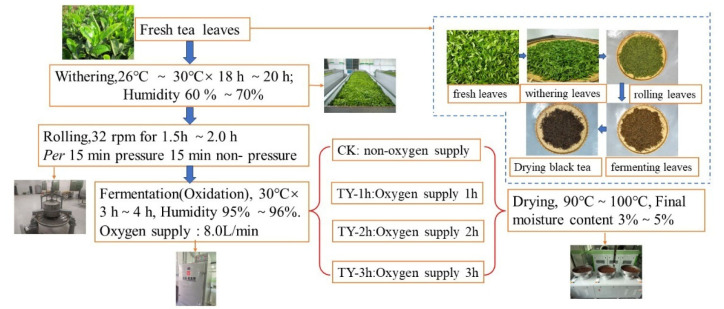
The schematic diagram of black tea processing. Note: CK—normal fermentation, non-oxygen supply. TY-1h—increased oxygen treatment 1 h. TY-2h—increased oxygen treatment 2 h. TY-3h—increased oxygen treatment 3 h. The same as below.

**Figure 2 foods-14-02736-f002:**
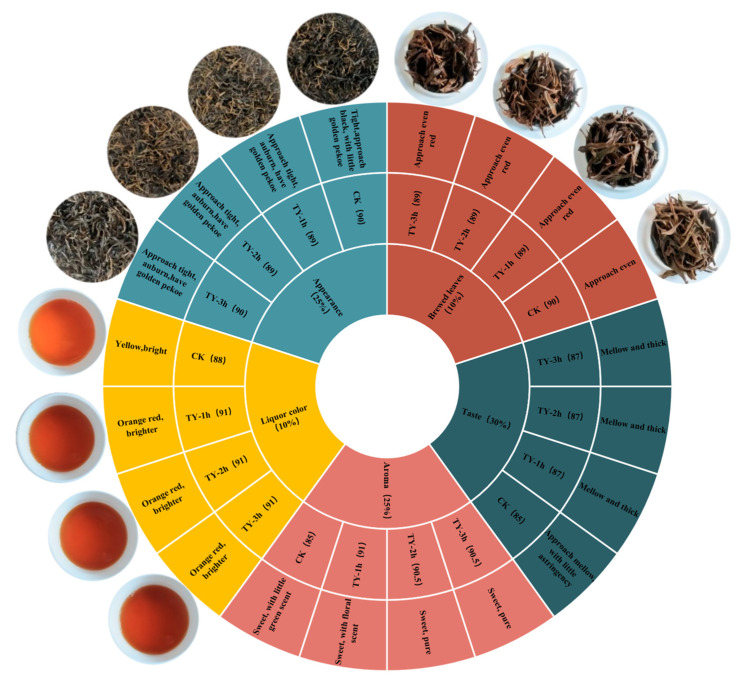
Sensory evaluation wheel of four black teas subjected to different treatments. Numbers represent sensory evaluation scores.

**Figure 3 foods-14-02736-f003:**
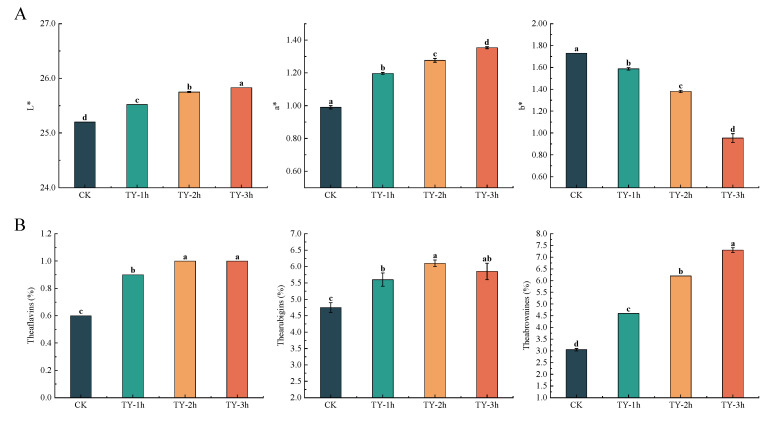
Comparison of liquor color attributes and tea pigment content among four black teas subjected to different treatments. (**A**) Liquor color attributes; (**B**) tea pigment content. Data are expressed as mean ± SD from at least three independent experiments. Bars labeled with different letters differ significantly at *p* < 0.05.

**Figure 4 foods-14-02736-f004:**
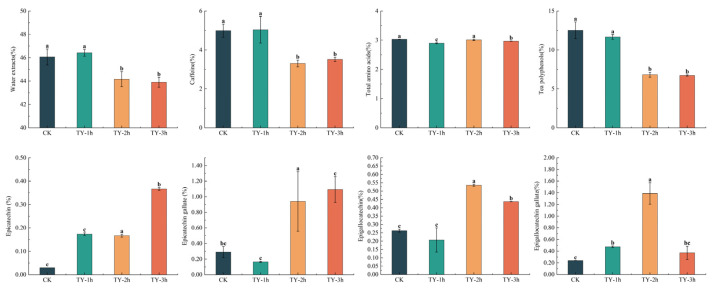
Comparison of general biochemical components among four black teas subjected to different treatments. Data are expressed as mean ± SD from at least three independent experiments. Bars labeled with different letters differ significantly at *p* < 0.05.

**Figure 5 foods-14-02736-f005:**
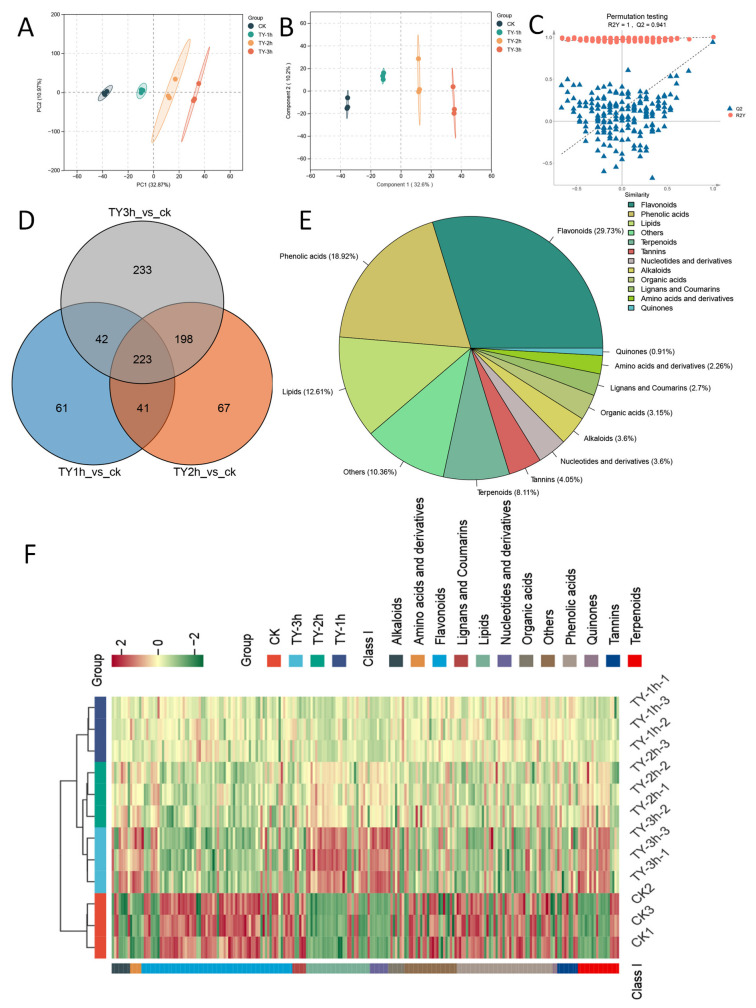
Metabolomics analysis of non-volatile metabolites at different treatments. (**A**) PCA; (**B**) OPLS-DA; (**C**) OPLS-DA model validation; (**D**) number of differential metabolites across CK and different oxygen supply time; (**E**) classification of differential metabolites among different comparison groups; (**F**) heatmap of the relative abundance of differential metabolites, with data standardized within rows and clustered in columns.

**Figure 6 foods-14-02736-f006:**
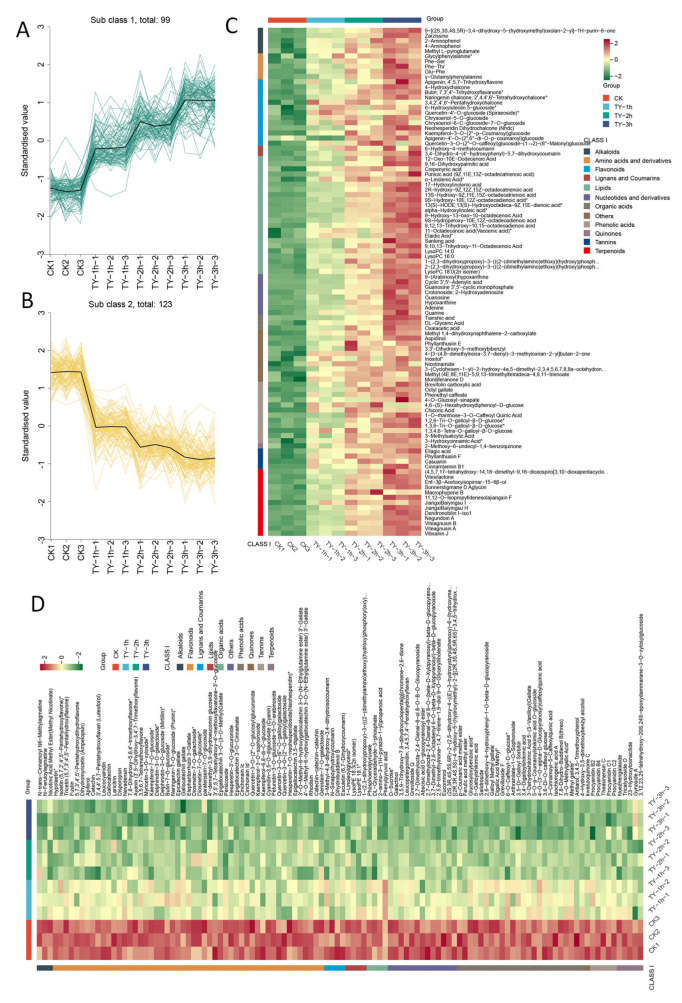
K-means clustering and abundance dynamics of differentially expressed non-volatile metabolites. (**A**) K-means clustering 1 and (**B**) clustering 2; (**C**) heatmap of upregulated DEMs; (**D**) heatmap of downregulated DEMs. Green and red segments denote the low and high abundance of metabolites, respectively.

**Figure 7 foods-14-02736-f007:**
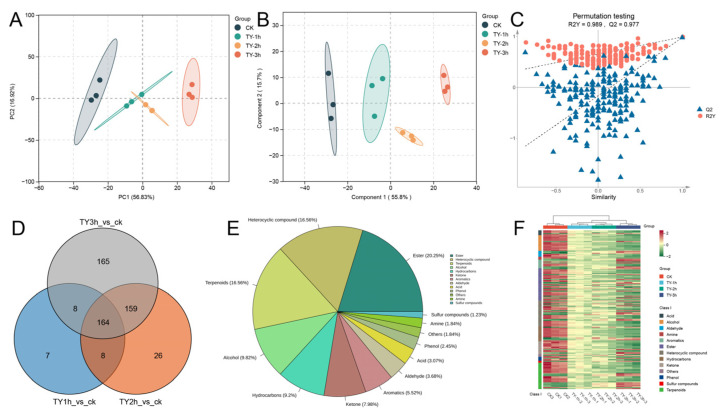
Volatile profiling across oxygen exposure durations. (**A**) PCA score plot; (**B**) OPLS-DA projection; (**C**) diagnostic validation of OPLS-DA model through permutation testing; (**D**) quantitative distribution of differential metabolites between experimental groups and control (CK); (**E**) taxonomic classification of shared differential metabolites across comparison groups; (**F**) hierarchically clustered heatmap depicting normalized relative abundance of consensus differential metabolites, with row-wise standardization and column-based clustering. Chromatic intensity reflects metabolite abundance gradients (green: reduced concentrations; red: elevated concentrations).

**Figure 8 foods-14-02736-f008:**
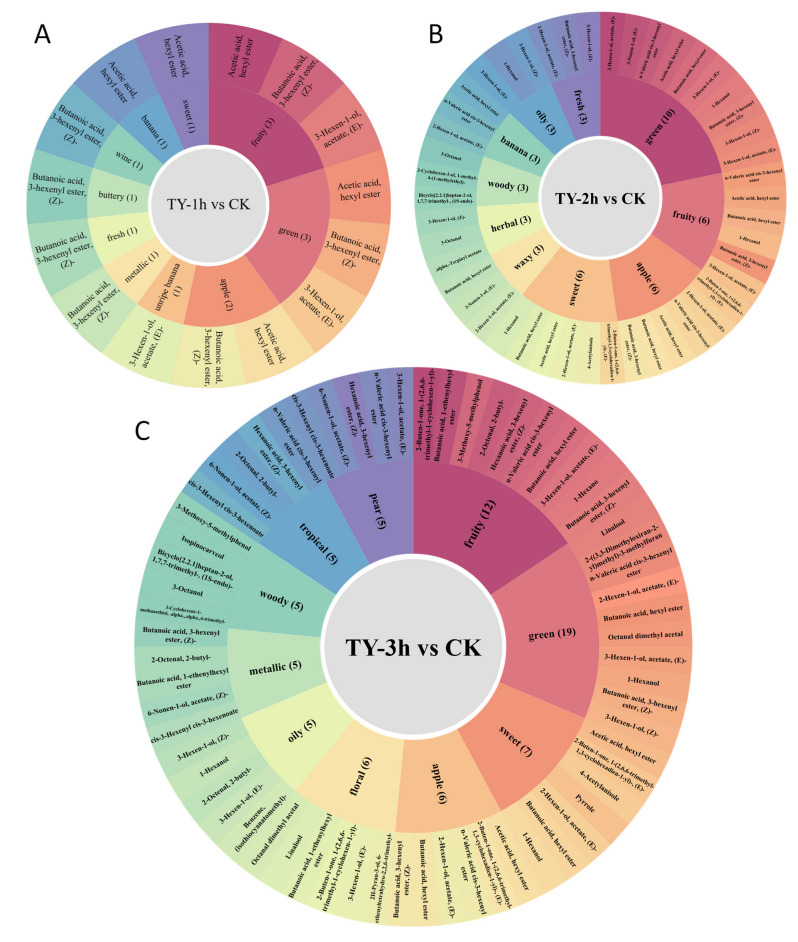
Temporal modulation of aromatic signatures in different treatments. (Presenting the top 10 differential volatile metabolites with the highest VIP values). (**A**) Initial phase oxygenation (TY-1h); (**B**) intermediate maturation stage (TY-2h); (**C**) prolonged aromatic development (TY-3h).

**Figure 9 foods-14-02736-f009:**
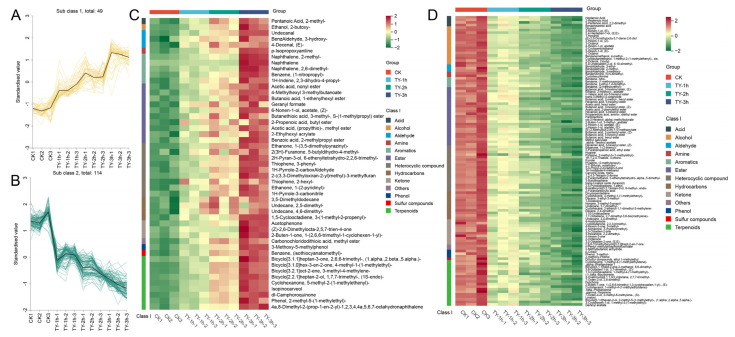
K-means clustering and abundance dynamics of differentially expressed volatile metabolites. (**A**) K-means clustering 1 and (**B**) clustering 2; (**C**) heatmap of up regulated DEMs; (**D**) heatmap of down regulated DEMs. Green and red segments denote the low and high abundance of metabolites, respectively.

## Data Availability

The original contributions presented in this study are included in this article/Appendix A; further inquiries can be directed to the corresponding authors.

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
