# Peer review of "Increased Oxygen Treatment in the Fermentation Process Improves the Taste and Liquor Color Qualities of Black Tea"

_foods, 2025, doi:10.3390/foods14152736_

Round 1
Reviewer 1 Report
Comments and Suggestions for Authors
The manuscript describes the investigation on the impact of oxygen supplementation during tea fermentation. This is an interesting experiment and would be of interest to the broader community. And while the paper is written in clear English there are serious issues with the presentation that cause serious issues with the paper.
Primarily, there are significant experimental details that are not included.
And in several section, more substantial referencing is needed.
In addition, here as several specific comments.
Line 161 & 172: The manuscript has the word "black" when I think the authors meant "blank"
The manuscript references Table 1 but I do not see the table included.
Section 3.3 this section need to be more completely referenced. A lot of statements here with no literature support.
In Figure 4 and in later discussion, the amount of caffeine is discussed and shown, however, the determination of caffeine is not included in any of the methods.
In section 3.4: the authors describe the PCA analysis which is not mentioned in the methods nor are any of the other statistical measure used in this section.
Figure 6 is discussed after figure 7 in the manuscript
Line 445: the incorrect reference style is used.
Author Response
|
3. Point-by-point response to Comments and Suggestions for Authors |
|
Comments 1: Primarily, there are significant experimental details that are not included. |
|
Response 1: Thank you for pointing this out. We agree with this comment. Therefore, We have already supplemented the details in the method section. |
|
Comments 2: And in several section, more substantial referencing is needed. Line 445: incorrect reference style is used. |
|
Response 2: Agree. We reviewed and improved the referening style and added substantial references for each statement in the manuscript. Changes were showed in the manuscript. |
|
Comments 3: The manuscript references Table 1 but I do not see the table included. |
|
Response 3: Agree. The correct referencing should be "Figure 2" instead of "Table 1". We replaced Table 1 with Figure 2 for more straightforward illustrate sensory evaluation result. |
|
Comments 4: Section 3.3 this section need to be more completely referenced. A lot of statements here with no literature support. |
|
Response 4: Agree. We have reviewed and added substantive references for each statement in the manuscript. |
|
Comments 5: In section 3.4: the authors describe the PCA analysis which is not mentioned in the methods nor are any of the other statistical measure used in this section. |
|
Response 5: Agree. Thank you for pointing out that, this part of the content has been supplemented to 2.9 Statistical Analysis. |
|
Comments 6: Figure 6 is discussed after figure 7 in the manuscript |
|
Response 6: Agree. Thanks for pointing this out, the first referencing of Figure 7 should be Figure 6, we revised in the manuscript. |
|
Comments 7: In Figure 4 and in later discussion, the amount of caffeine is discussed and shown, however, the determination of caffeine is not included in any of the methods. |
|
Response 7: Agree. The detection of caffeine is the same as that of catechin, both by HPLC, and caffeine has been added to the Method section. |
|
4. Response to Comments on the Quality of English Language |
|
Point 1: Line 161 & 172: The manuscript has the word "black" when I think the authors meant "blank" |
|
Response 1: Thanks for pointing this out, we checked the corresponding sentences, it's a typo and should be "blank" instead of "black". It has been revised. |

Reviewer 2 Report
Comments and Suggestions for Authors
The manuscript is noteworthy because it evaluates the effect of different oxygen concentrations on the color and flavor of black tea, which is of interest to professionals in the field.
Nevertheless, the authors need to make a series of changes to improve the quality of the manuscript.
- L:32 Remove the hyphen from the word antihypertension, and in L:36, remove the hyphen from the word polyphenolic
- Follow the journal's format for citations and references ( L:43, 46, 50)
- Capitalize National Standard in L: 142 and 149
- Please rewrite the indicated sentences (L:151, 164-165, 237-240
- Change foline-phenol to Folin-Ciocalteu reagent
- L:177 How was the systematic analysis performed? Please explain
- L:254 Put a space between than and 99.99%
- L:269- Why was the Fisher Least Significant Difference (LSD) test used instead of the Tukey test? Justify this.
- All figures are tiny and difficult to see. Consider enlarging them and sending some as supplementary material so that they can be seen better.
- Where two or three quantities appear with percentages, only place the percentage symbol on the final figure. Correct this throughout the manuscript.
- Table 1 is not included.
- Excessive use of capital letters throughout the manuscript; please do not overuse them.
- L:427 Change "gustatory analysis" to "taste analysis."
- Avoid the use of personal pronouns such as "our," "your," etc. (L:487,608)
Author Response
|
Comments 1: Follow the journal's format for citations and references ( L:43, 46, 50) |
|
Response 1: We agree with this comment. |
|
Comments 2: L:177 How was the systematic analysis performed? Please explain |
|
Response 2: Thanks for your question. We replaced the systematic analysis with the spectrophotometry analysis. It might be more clear for the readers. |
|
Comments 3: L:269- Why was the Fisher Least Significant Difference (LSD) test used instead of the Tukey test? Justify this. |
|
Response 3: In this study, given the limited sample size and the emphasis on detecting significant intergroup differences, the least significant difference (LSD) method was employed to enhance the sensitivity of our analytical results. This approach prioritizes statistical power in identifying variations between groups under constrained experimental conditions. |
|
Comments 4: All figures are tiny and difficult to see. Consider enlarging them and sending some as supplementary material so that they can be seen better. |
|
Response 4: Agree. We will upload each picture separately in the system to make it clearly visible. Thank you. |
|
Comments 5: Table 1 is not included. |
|
Response 5: Agree. The correct referencing should be "Figure 2" instead of "Table 1". We replaced Table 1 with Figure 2 for more straightforward illustrate sensory evaluation result. |
|
4. Response to Comments on the Quality of English Language |
|
Point 1: L:32 Remove the hyphen from the word antihypertension, and in L:36, remove the hyphen from the word polyphenolic |
|
Response 1: Thanks for your advice, we remove the two hyphens and checked similar issues across the manuscript. |
|
Point 2: Capitalize National Standard in L: 142 and 149 |
|
Response 2: Agree, we revised the formatting. |
|
Point 3: Please rewrite the indicated sentences (L:151, 164-165, 237-240 |
|
Response 3: Thank you for pointing out these points. We have rewritten the sentence to make it convenient for reading. |
|
Point 4: Change foline-phenol to Folin-Ciocalteu reagent |
|
Response 4: Thanks for your advice, we revised it. |
|
Point 5: L:254 Put a space between than and 99.99% |
|
Response 5: Thanks for your advice, we revised it. |
|
Point 6: Where two or three quantities appear with percentages, only place the percentage symbol on the final figure. Correct this throughout the manuscript. |
|
Response 6: Thanks for your advice, we revised it throughout the manuscript. |
|
Point 7: Excessive use of capital letters throughout the manuscript; please do not overuse them. |
|
Response 7: Thanks for your advice, we revised it. |
|
Point 8: L:427 Change "gustatory analysis" to "taste analysis." |
|
Response 8: Thanks for your advice, we revised it through out the manuscript. |
|
Point 9: Avoid the use of personal pronouns such as "our," "your," etc. (L:487,608) |
|
Response 9: Thanks for your advice, we could not find the mentioned issue at the correspondent lines (L:487,608), but we checked through out the manuscript and depressed the use of personal pronouns. |
|
5. Additional clarifications |
|
[Here, mention any other clarifications you would like to provide to the journal editor/reviewer.] |

Round 2
Reviewer 1 Report
Comments and Suggestions for Authors
The manuscript describes the impact of supplemental oxygen during the fermentation process of tea. The authors have greatly improve the manuscript and clearly addressed all the previous concerns. As such the manuscript is greatly improved and I see no additional reason not accept the paper at this time.
Reviewer 2 Report
Comments and Suggestions for Authors
The authors have made enough changes for the paper to be accepted